# Sudden, local temperature increase above the continental slope in the Southern Weddell Sea, Antarctica

Elin Darelius[1], Vår Dundas[1], Markus Janout[2], and Sandra Tippenhauer[2]

[1]Geophysical Institute, University of Bergen and the Bjerknes Centre for Climate Research, Bergen, Norway

[2]Alfred-Wegener- Institute Helmholtz Centre for Polar and Marine Research, Bremerhaven, Germany

**Correspondence:** E. Darelius (elin.darelius@uib.no)

**Abstract.** Around most of Antarctica, the Circumpolar Deep Water (CDW) shows a warming trend. At the same time, the thermocline is shoaling, thereby increasing the potential for CDW to enter the shallow continental shelves and ultimately increase basal melt in the ice shelf cavities that line the coast. Similar trends, on the order of 0.05 °C and 30 m per decade, have been observed in the Warm Deep Water (WDW), the slightly cooled CDW derivative found at depth in the Weddell Sea.
Here we report on a sudden, local increase in the temperature maximum of the WDW above the continental slope north of the Filchner Trough (74°S, 25-40°W,), a region identified as a hotspot for both Antarctic bottom water formation (AABW) and potential changes in the flow of WDW towards the large Filchner-Ronne Ice Shelf.

New Conductivity-Temperature-Depth profiles, obtained in summer 2021, and recent (2017-2021) mooring records show that the temperature of the warm water core increased by about 0.1°C over the upper part of the slope (700 - 2750 m depth) compared with historical (1973-2018) measurements. The temperature increase occured relatively suddenly in late 2019 and was accompanied by an unprecedented (in observations) freshening of the overlying Winter Water. The AABW descending down the continental slope from Filchner Trough is sourced by dense Ice Shelf Water and consists to a large degree (60%) of entrained WDW. The observed temperature increase can hence be expected to imprint directly on deep water properties, increasing the temperature of newly produced bottom water (by up to 0.06°C) and reducing its density.

## 1 Introduction

The Antarctic ice sheet stores a large amount of freshwater with the potential to significantly increase the global sea level. The continent is fringed by floating ice shelves, buttressing and slowing down the ice streams into the surrounding ocean. However, in recent decades the ice shelves have been losing mass at an accelerated rate (Paolo et al., 2015). The largest ice shelf melt rates are found in the Amundsen Sea, where the continental shelf and the ice shelf cavities are flooded by warm Circumpolar Deep Water (CDW). The southward oceanic heat fluxes are further amplified by the continuing warming of the

Southern Ocean (Schmidtko et al., 2014) and are likely to result in meltwater that can change the oceanic density structure. For example, the freshening observed throughout the last decades in the Ross Sea that may ultimately affect the dense water formation and thus impact the global ocean circulation is likely a consequence of increasing Amundsen Sea meltwater input upstream (Jacobs et al., 2022). The Filchner-Ronne Ice Shelf (FRIS) is the earth's largest ice shelf by volume and presently experiences comparatively moderate basal melt rates (Rignot et al., 2013). While the Weddell Sea, in concert with the global ocean, also shows multi-decadal warming and freshening (Strass et al., 2020), FRIS is protected from the oceanic heat by the vast cold southern Weddell Sea continental shelf. This shelf is an important sea ice formation region, which results in the production of High Salinity Shelf Water (HSSW), a near-freezing dense water mass, that enters the cavity of the Ronne Ice Shelf. The HSSW then interacts with the base of the ice shelf at several hundred meters depth, where the local freezing point is a few tenths of a degree lower than at the surface. This leads to ice shelf melt and transformation of the HSSW into slightly fresher and colder Ice Shelf Water (ISW). ISW then exits across the front of the Filchner Ice Shelf into the Filchner Trough (FT), and thus completes an anti-cyclonic circulation. Both HSSW and ISW are exported down the continental slope and form the precursors of Antarctic Bottom Water (AABW, Foldvik et al., 2004), which is an important part of the global ocean circulation. Furthermore, these cold and dense waters dominate the water masses on the continental shelf and block any large-scale presence of Warm Deep Water (WDW, $0 < T < 0.9^{+}circ$C, Gammelsrød et al., 1994) on the continental shelf. WDW is the slightly colder Weddell Sea equivalent of Circumpolar Deep Water and occupies the subsurface Weddell Sea basin north of the continental slope. The WDW lies below a layer of Winter Water (WW), with temperatures at or close to the surface freezing point. The WDW-WW interface, which represents the thermo- and pycnocline, deepens towards the continental shelf break, where it forms the Antarctic Slope Front and the associated Antarctic Slope Current, which flows westward along the shelf break. Seasonal changes in wind forcing and stratification (Hattermann, 2018) cause the thermocline to deepen during winter and relax during summer, leading to a seasonal inflow of WDW, or its modified form (mWDW), on the continental shelf east of the FT (Årthun et al., 2012; Ryan et al., 2017) that may reach the Filchner ice front (Darelius et al., 2016). MWDW also enters the continental shelf in a trough further west (Nicholls et al., 2008), where it, strongly modified ($T = -1.4$C), reaches the Ronne Ice Shelf cavity (Foldvik et al., 1980; Janout et al., 2021; Davis et al., 2022). On the large scale, however, dense and cold waters dominate the southern Weddell Sea shelf, although numerical modeling experiments project changing conditions toward the end of this century (Hellmer et al., 2012). In these projections, strongly enhanced mWDW inflow over the eastern continental shelf drastically increases basal melt rates underneath FRIS. Regional modeling efforts, however, suggest that the system is relatively resistant to changes in wind forcing and thermocline depth (Daae et al., 2019), while FRIS melt rates respond linearly to changes in the salinity changes of the Antarctic Slope Current (Bull et al., 2021).

Here we present new observations: Conductivity-Temperature-Depth (CTD) profiles collected during the COSMUS expedition on RV Polarstern in February-March 2021 and oceanographic records from moorings recovered during the same cruise. The data show a sudden, local warming of the WDW above the upper part of the continental slope in the FT region towards the end of 2019 and an apparent freshening. Given the importance of the region for the AABW production (Foldvik et al., 2004), the possibility for more than an order of magnitude increase in melt rates (Hellmer et al., 2012), and for a "regime shift" in the

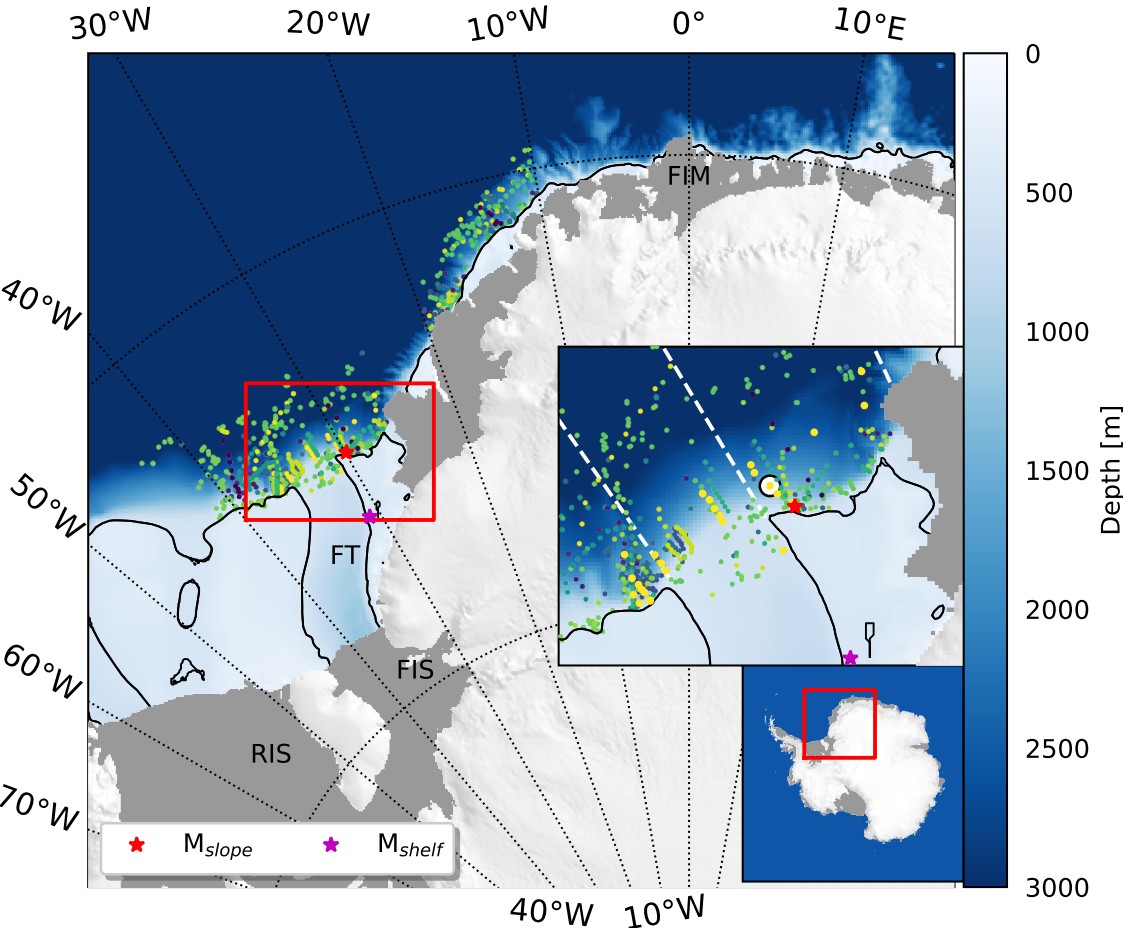

**Figure 1.** Map showing the bathymetry (Bedmap2, blue shading) of the south-western Weddell Sea (red box in lower inset) and the location of CTD-profiles included in the study (color-coded in time using the same color bar as in Fig. 3). Floating ice shelves are marked in grey and the 500-m isobath is shown in black. Mooring positions are indicated with colored stars according to the legend. FT: Filchner Trough, FIS: Filchner Ice Shelf, RIS: Ronne Ice Shelf, FIM: Fimbul Ice Shelf. The upper inset shows a zoom-in on the study area (red box), with CTD – profiles from PS124 in 2021 highlighted (larger, yellow dots). The position of the profile shown in Fig. 2 is highlighted with a white and black circle. The longitudes used to divide the slope north of the FT (25W, 31W, and 35W) are indicated with white dashed lines in the inset (45W is west of the area shown).

southern Weddell Sea (Hellmer et al., 2017; Daae et al., 2020; Bull et al., 2021), the observed changes deserve our attention as a potential "canary in the coal mine".

## 2 Data and methods

### 2.1 Conductivity-Temperature-Depth (CTD) profiles

More than 1000 historical CTD profiles (of which over 600 were collected by ship, over 200 by instrumented Weddell Seals (Treasure et al., 2017) and the rest by profiling Argo floats) from the continental slope in the southern Weddell Sea (between 45-10°W, 500-3500 m depth) were identified in publicly available databases. The majority of the seal profiles available from the slope area were excluded since the seals rarely dive deep enough over the slope to capture the temperature maximum, and not all of the shallower stations displayed temperatures high enough to be considered. In the end, 669 profiles were included in total. The historical data set spans the time period between 1973 and 2020, where the majority of the profiles are obtained during the Austral summer (January-February), see Fig. A1. The bottom depth at the profile location was interpolated from BEDMAP2 (Fretwell et al., 2013).

The historical data set is complemented by 25 profiles collected between 11 February - 15 March 2021 during PS124 onboard RV Polarstern as part of the COSMUS (Continental Shelf Multidisciplinary Flux Study)- Expedition (Hellmer and Holtappels, 2021). The profiles were collected using a standard CTD/Rosette SeaBird SBE911plus system, equipped with double sensors for temperature, salinity, and oxygen and one sensor each for pressure, substance fluorescence chlorophyll $a$, and beam transmission. Sensors were calibrated by the manufacturer before and after the cruise. Additionally, water samples were taken (using 24 12-liter Niskin bottles attached to the rosette) and measured with the Optimare Precision Salinometer (OPS) for in-situ calibration of the conductivity sensor.

The location of the available CTD profiles is shown in Fig. 1, and the analysis is carried out over four different sections of the slope; the area northwest of the FT (35-45°W), the area north of the FT (31-35°W), the area northeast of the Trough (25-31°W), and the slope upstream of the FT(10-20°W).

Temperature and salinity are reported in Conservative Temperature, $\Theta$, and absolute salinity $S_A$ (McDougall, 2011; IOC et al., 2010).

#### 2.1.1 Mooring records

The CTD data are complemented by records from a mooring located on the continental slope east of the FT; M3 at 750 m depth (Fig. 1). The mooring will in the following be referred to as $M_{slope}$. The mooring was deployed during cruise JR16004 in 2017 and recovered during PS124 in 2021, providing up to four-year-long records of temperature, salinity, and current speeds. In addition, one-year-long historical records from the $M_{slope}$ position in 2009-2010 are included (Jensen et al., 2013; Semper and Darelius, 2017; Fer, 2016). To investigate the potential on-shelf propagation of the signal observed above the slope, we include temperature records from mooring $M_{shelf}$ (referred to as $M_{30W}$ by Ryan et al., 2017, 2020), deployed at 450 m depth on the continental shelf at about 76°S . The historical records (2014-2018, Schröder et al., 2017, 2019) are extended to 2021 (Janout et al., 2022). Mooring positions are indicated in Fig. 1, and details about the mooring are given in Table 1. We estimate the eddy kinetic energy (EKE) in four frequency bands: around 12h (semi-diurnal tides, B12), around 24h (dirunal tides, B24), around

35h (continental shelf waves, B35, Jensen et al., 2013), and 2 days to 14 hours following Jensen et al. (2013). The frequency bands are marked in Fig. 6e.

**Table 1.** Information about the moorings included in the study. Mooring locations are shown in Fig. 1.

|  | Lon | Lat | Bottom depth [m] | Deployment periods |
|---|---|---|---|---|
| $M_{slope}$ | 29.91°W | 74.55°S | 750 | 2009, 2017-2021 |
| $M_{shelf}$ | 30.99°W | 76.05°S | 450 | 2014-2016, 2016-2018, 2018-2021 |

## 3  Results

The CTD profile obtained above the continental slope at 1500 m depth north of the FT (25-31 °W) in February 2021 (Fig. 2) shows the water mass layering typical for the region and the season: a shallow (35 m), relatively fresh ($S_A$<34 g kg$^{-3}$, not shown), and solar heated surface layer overlaying a quasi-homogeneous layer of WW with $\Theta$ just above the freezing point ($\Theta_{FP}$) and $S_A \simeq$34.5 g kg$^{-3}$. Below the WW-layer, which at this station extends down to about 400 m depth, the temperature increases rapidly towards a maximum of $\Theta$=0.76°C at about 850 m depth. This is the core of the WDW. Below the WDW core, the temperature decreases towards that of Weddell Sea Deep and Bottom Water, which is found at greater depths in the Weddell basin.

We note in Fig. 2a), that the maximum temperature of the WDW core in 2021 was more than 0.1°C warmer than the core temperature of any previous $\Theta$-profile from this part of the slope. Similarly, anomalously high WDW core temperatures are observed all over the upper part of the slope, both east, north, and west of the FT (Fig. 3a-c). While temperatures around 0.8°C are regularly observed above the deeper part of the slope, only the profiles from 2021 display temperatures in this range for bottom depths shallower than about 2000 m. The signal is clearest east of the FT (Fig. 5a), where one has to move even further down the slope to find historical observations of water that is around 0.8°C. Moving further east, e.g. to the steep continental slope between 10-20°W, the core temperature above the slope is generally higher, also above the upper part of the slope (Fig. 3d). Unfortunately, there are no data from this region in 2021. While the profiles from 2021 were collected relatively late (8 Feb - 15 March) in the summer season, there is no indication in the data that the time of sampling can explain the high temperatures observed (Fig. A2).

The high temperatures of the WDW core are accompanied by relatively high salinities in the WDW depth range. Still, the most noteworthy feature of the salinity profile from 2021 in Fig. 2b is the low salinity of the WW layer. The WW is fresher than any previously observed salinity and about 0.025 g kg$^{-1}$ fresher than the mean value. Low values of WW-layer salinities were observed at all stations east of the FT in 2021 (Fig. 4), but note that the variability in the observed values is relatively high. While it is beyond the scope of this paper, the freshening appears to align with a general freshening trend of the WW

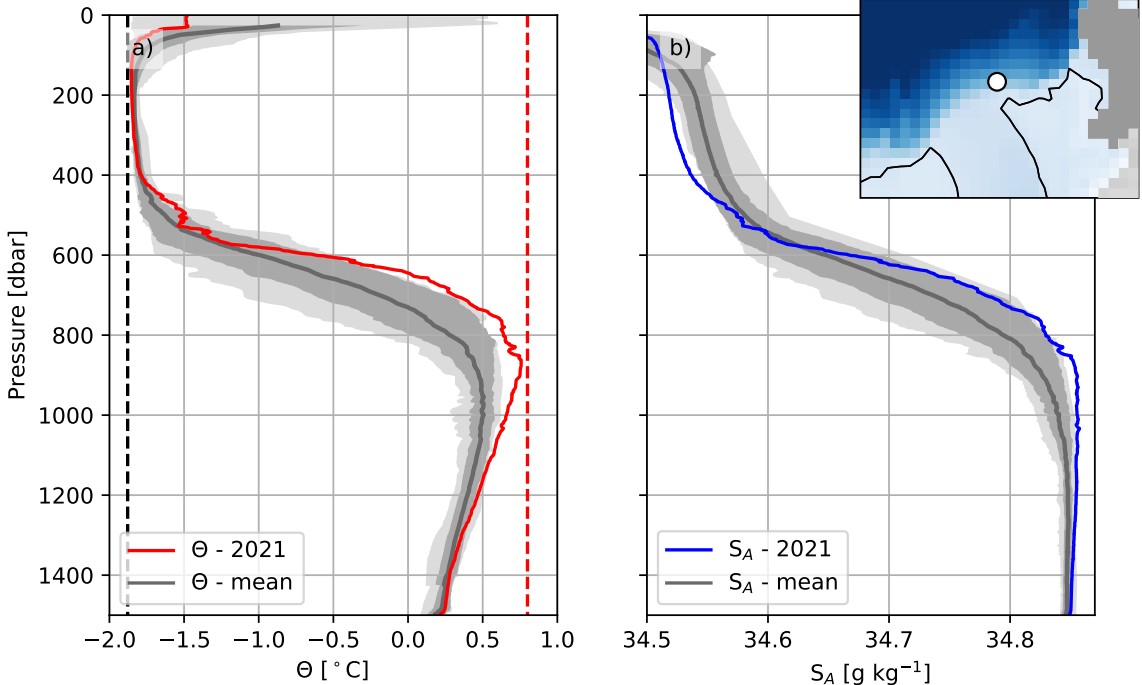

**Figure 2.** Profiles of a) Conservative Temperature ($\Theta$, red line) and b) Absolute Salinity ($S_A$, blue line) on February 13th, 2021, on the continental slope east of the FT above the 1515 m isobath (white circle in map in the inset). Mean profiles of $\Theta$ and $S_A$ (gray lines) from 13 stations occupied between the the 1300 m and the 1700 m isobath east of the FT (25-31°W) in the period 1973-2017 are included for comparison. The dark shaded area denotes $\pm$ one standard deviation, and the light, shaded area is the range of values observed at a given depth. The freezing temperature ($\Theta_{FP}$, dashed black line) and $\Theta$=0.8°C (dashed red line) are highlighted in (a).

layer in the FT region. The WW-layer typically erodes towards the west due to thermocline shoaling (see Fig. 5 Darelius et al., 2023) and mixing, and while the salinities at 250 m west of the FT were in the lower end of the observed range (Fig. 4a), the signal is strongest north and east of the FT. The mooring data suggest a pronounced seasonality in the 300 m salinity, with a decrease in salinities occurring in April (not shown), potentially as a result of an advected freshwater anomaly from the east (Graham et al., 2013), but the CTD profiles from 2021 were obtained in mid-February to mid-March, and there is no indication in the mooring records of an early freshening in 2021.

Fig. 2 suggests that the 2021 profiles, in addition to a higher core temperature, also display a shallower thermocline, although this is not supported by all observations, especially not north of and west of the FT (Fig. 5). It is, however, not straightforward to interpret potential interannual variability in thermocline depth from the scattered CTD profiles, since the thermocline depth varies spatially and temporally, both on seasonal (Årthun et al., 2012; Semper and Darelius, 2017) and shorter time scales

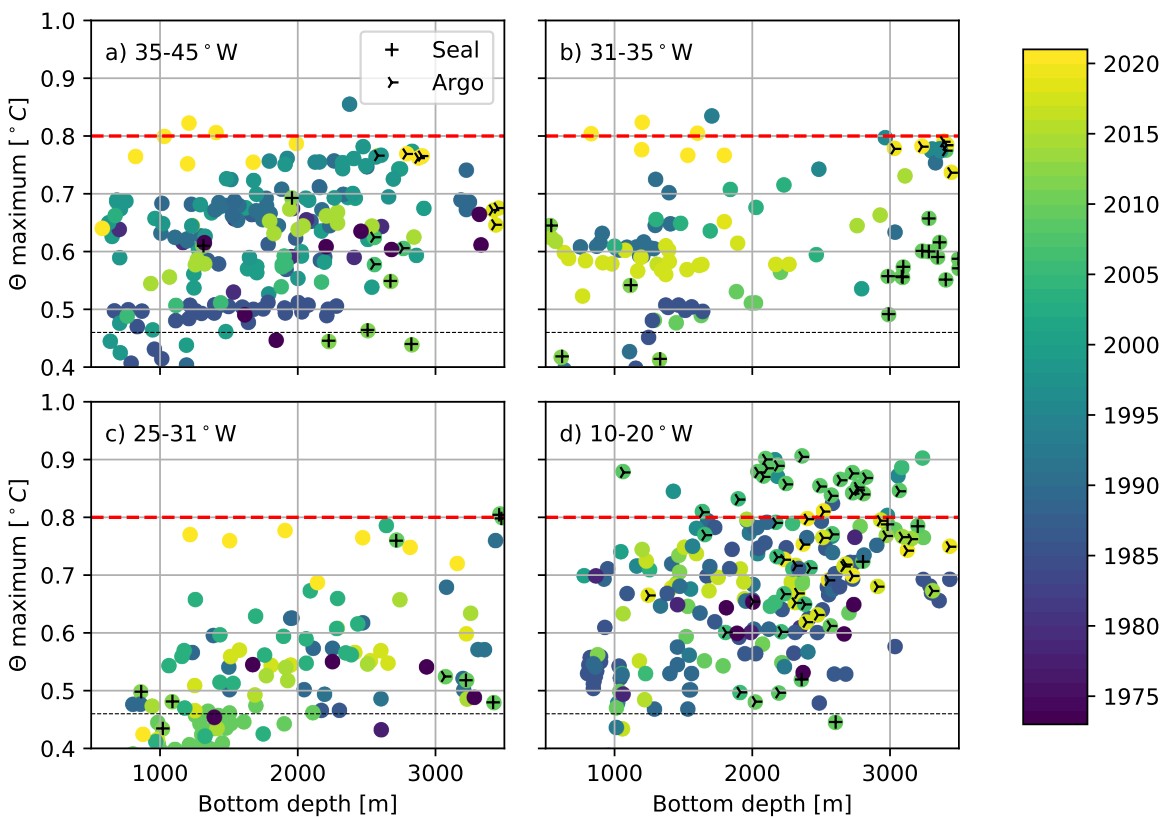

**Figure 3.** Maximum conservative temperature within the WDW-core from CTD-profiles obtained above the continental slope a) northwest of FT (35-45°W) b) north of the FT (31-35°W) c) northeast of the FT (25-31°W) and d) upstream of the FT region (10-20°W) as a function of bottom depth. The data points are color-coded with respect to time, and profiles obtained by seals or profiling floats are marked according to the legend. Note that seal profiles rarely cover the temperature maximum; the temperature shown is the highest temperature observed during a dive. The red dashed line marks 0.8°C and is included in all panels (and in subsequent figures when relevant) to facilitate comparison. The black line is the average two-week maximum temperature from the mooring records (see Fig. 6a-d and the text for an explanation).

(Darelius et al., 2009; Middleton et al., 1982; Jensen et al., 2013). The shoaling of the thermocline west of the FT discussed by Darelius et al. (2023) is evident in Fig. 5.

The results from the CTD profiles (Fig. 2) are corroborated by mooring records obtained from the slope (Fig. 6), where we note a sudden change in the temperature record towards the end of 2019 (Fig. 6b,d). Since the temperature variability is high on daily time scales (due to shelf waves advecting the thermocline across the slope; Jensen et al., 2013; Semper and Darelius, 2017), we consider the mean and maximum temperature in two-week-long windows (Fig. 6a-d) and refer to them as $\mathrm{mean}_{2w}$

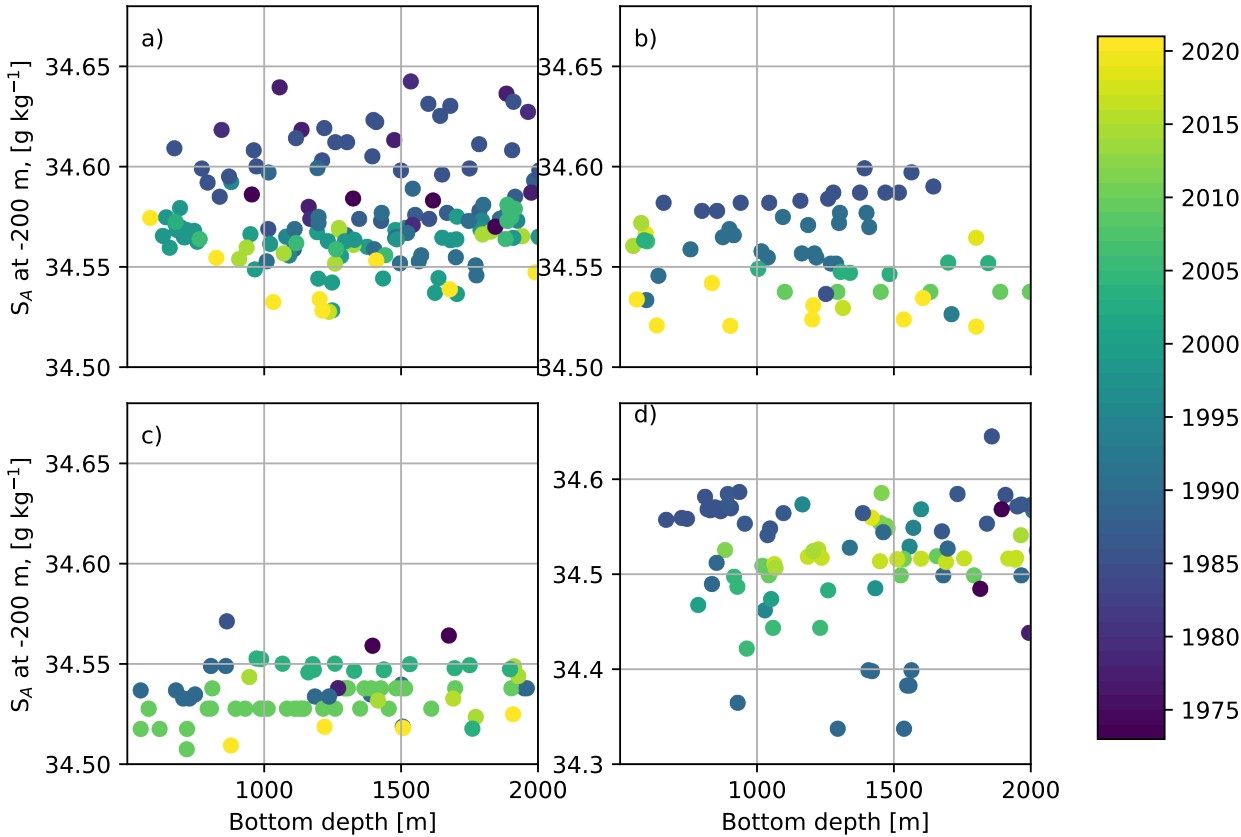

**Figure 4.** Absolute salinity at 250 m depth as a function of bottom depth a) northwest of FT (35-45°W) b) north of the FT (31-35°W) c) northeast of the FT (25-31°W) and d) upstream of the FT region (10-20°W). Profiles obtained by seals or profiling floats are excluded from the analysis. Note that the scale in panel (d) differs from that of the other panels.

and maximum$_{2w}$ temperature, respectively. The mean$_{2w}$ temperature records from M$_{slope}$ show the seasonality typical for the region during the first part of the record, with a late summer maximum when the mooring is surrounded by mWDW and WDW, and a minimum in winter when WW is observed to reach down to the bottom (Semper and Darelius, 2017, Fig. 6a-d). In November 2019, the maximum$_{2w}$ temperature increases by about 0.1°C, and for the rest of the record, it never decreases below the average maximum$_{2w}$ temperature, not even in winter (dashed grey line, Fig. 6b). The seasonal signal in 2020 is similarly disrupted; throughout much of the winter, the mean$_{2w}$ temperature is above the mean seasonal cycle, which is inferred from the 5-year long time series (red line in Fig. 6d). The absolute maximum$_{2w}$ temperature (0.76°C) occurs in January 2021 and is

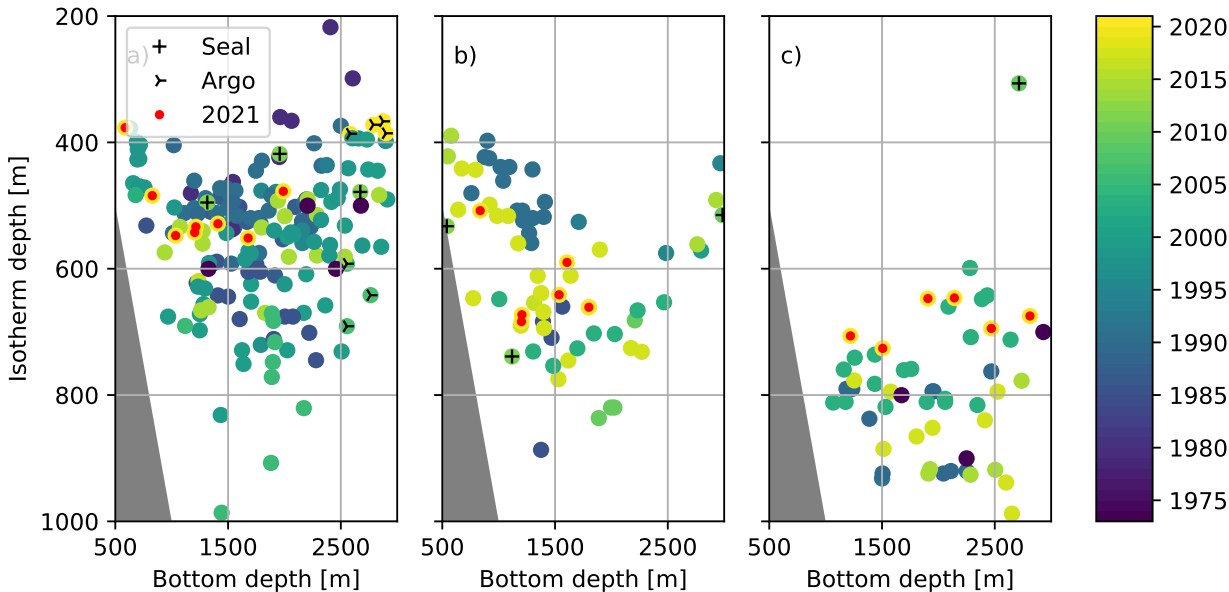

**Figure 5.** Scatter plot of the depth of the 0.5°C isotherm versus bottom depth, where the color code indicates the year a) northwest of the FT (35-45°W) b) north of the FT (31-35°W) and c) northeast of FT (25-31°W). Profiles obtained by instrumented seals, Argo floats, or in 2021 are marked according to the legend.

roughly 0.30°C higher than the average maximum$_{2w}$ temperature and 0.17°C warmer than the highest temperature observed prior to November 2019 (Fig. 6b).

Energetic shelf waves traveling along the upper part of the slope can potentially affect the temperature observed at M$_{slope}$ by moving the thermocline up and down and by increasing mixing in the bottom boundary layer. The EKE associated with the diurnal shelf waves varies seasonally, with a maximum in summer, as changing hydrography and currents modify their dispersion relation and the possibility for resonance (Semper and Darelius, 2017). A similar seasonality (suggested by Jensen et al., 2013, based on one-year-long records) is apparent for shelf waves with a period of 35h in the M$_{slope}$ records (Fig. 6f). The mid-depth (250-500 m) EKE record (Fig. 6f) shows that i) the summertime peak in EKE is reduced in 2020 - 2021 (or already in 2019 for the 35h-band) and ii) the (smaller) wintertime peak in EKE associated with the diurnal tide (B24) is enhanced in 2020 (Fig. 6f, green line) when the mean$_{2w}$ temperature is above the seasonal average. From the tidal forcing alone, we would expect quasi-symmetric bi-annual peaks for the EKE related to diurnal tides (see e.g. Semper and Darelius, 2017, Fig. 6f).

M$_{shelf}$ that monitors the warm inflow towards the Filchner ice front (Ryan et al., 2017, 2020), on the other hand, does not show a warming signal after 2019. While the seasonal warm inflow (Årthun et al., 2012) in 2017 was "exceptionally warm and prolonged" (Ryan et al., 2020) and the inflow in 2018 even warmer (but shorter), the maximum temperature in 2020-2021 are

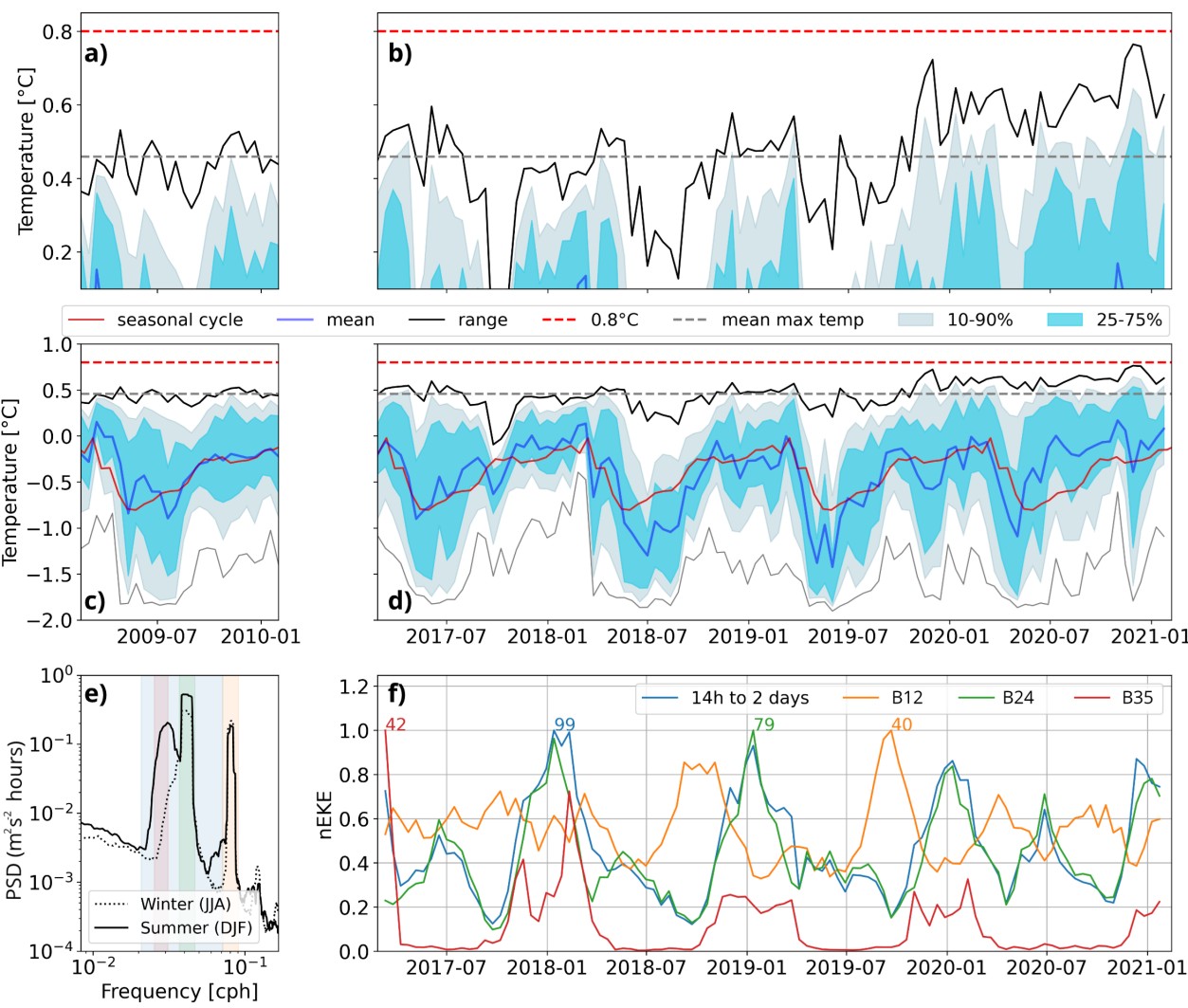

**Figure 6.** a-d) Time series of bottom temperature from $M_{slope}$ averaged over two-week-long windows, including maximum (black), minimum (grey), mean (blue), the 10-90 (filled, grey), and the 25-75 (filled, light blue) percentiles. The average maximum temperature (grey dashed line), $0.8°C$ (red dashed line), and the mean seasonal cycle based on the five-year-long records (red line) are indicated. a-b) is a zoom-in of the highest temperatures at $M_{slope}$. e) Frequency spectra of across-slope velocity (250-500 m depth) at $M_{slope}$ for winter (dotted line) and summer (black). The shading marks the B35 (red), B24 (green), B12 (orange), and 14 hours to 2 days (blue) frequency bands, color-coded to correspond with panel f). f) Time series of normalized vertical mid-range EKE (250-500 m depth) at mooring $M_{slope}$ in the same four frequency bands. The colored numbers show the maximum EKE value (in $cm^2s^{-2}$) for each frequency band.

closer to those observed prior to 2017. The disruption in the seasonal cycle observed towards the end of the record on the slope (6d) is not observed on the shelf.

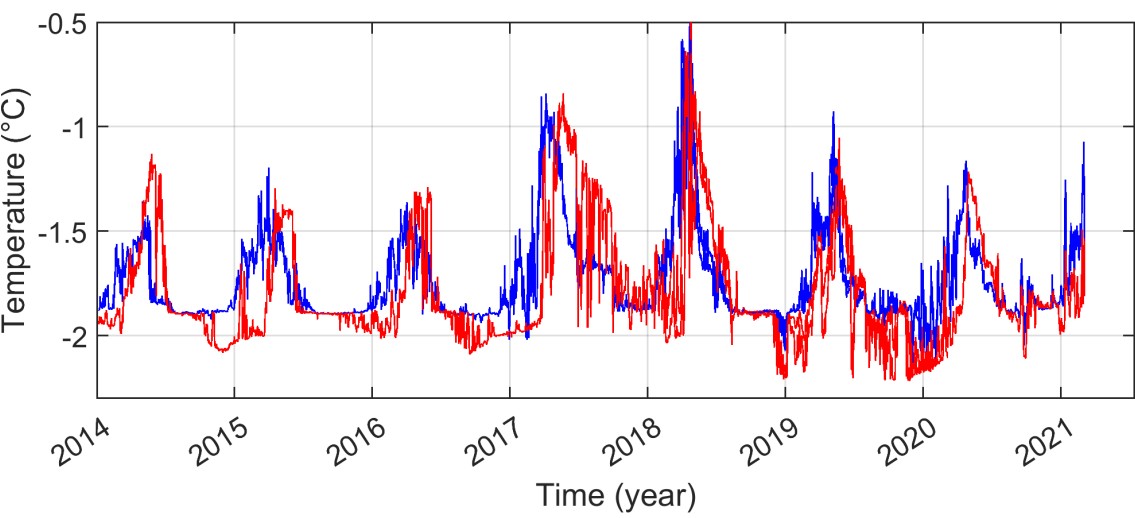

**Figure 7.** Time series of temperature at 340 m (blue) and 430 m (red) depth from mooring M$_{shelf}$ (A253) on the continental shelf at 76°S, east of the FT (Ryan et al., 2017, 2020).

## 4 Discussion

Observations from the southern Weddell Sea show a sudden, local warming of the WDW-core above the upper part of the continental slope in the region north of the FT (Fig. 2-3, 6a-b). CTD profiles from 2021 reveal that the temperature maximum was about 0.1°C warmer than previously observed. The changes are most pronounced east of the FT above the upper part of the slope (shallower than 2750 m). Mooring records suggest a change to have occurred during 2019, as the maximum temperature in two-week-long windows (maximum$_{2w}$ temperature) observed at the mooring from November 2019 and onward are markedly higher than before (Fig. 6b). During 2020, the seasonal signal in temperature (Semper and Darelius, 2017; Årthun

et al., 2012), which is evident in the early part of the record, is not as dominant: the mean temperature during two-week-long windows (mean$_{2w}$ temperature) is markedly higher during this winter compared to previous winters.

The temperature of the WDW above the slope is important in at least three ways; the mWDW enters the continental shelf where it, if it enters the FRIS cavity, can increase basal melt (Hellmer et al., 2012, 2017), mWDW entering the continental shelf acts as a precursor to HSSW formation (Nicholls et al., 2009), the WDW is one of the main components of the bottom water produced in the Weddell Sea, as it is entrained into the plumes of dense shelf waters that descend the continental slope (Foldvik et al., 2004). Therefore, the observed temperature increase has implications for (1) mWDW-inflow and potential basal melt, (2) HSSW production, and (3) bottom water production. These points will be further discussed below.

## 4.1 mWDW-inflow and potential basal melt beneath FRIS

MWDW is known to enter the continental shelf east of the FT seasonally (Årthun et al., 2012) and to flow southwards towards the Filchner Ice Shelf cavity (Ryan et al., 2017; Darelius et al., 2016). The temperature time series from M$_{shelf}$ (Fig. 7) accordingly show that the mooring is alternatingly surrounded by relatively warm mWDW and colder shelf waters/ISW. While the warm inflow was "exceptionally warm and prolonged" in 2017 (Ryan et al., 2020), the inflow in 2020 – when temperatures above the slope were increased and the seasonality reduced (Fig. 6) – was neither exceptionally warm nor long (Fig. 7). This suggests that processes and factors other than the WDW core temperature on the upper part of the slope determine the temperature, strength, and duration of the warm inflow and that the temperature increase would not directly influence basal melt rates.

## 4.2 HSSW-production

The mWDW on the upper part of the continental slope enters the continental shelf also in a depression west of the FT (Nicholls et al., 2008) where it is suggested to be the main source water for the production of HSSW in front of the Ronne Ice Shelf (Nicholls et al., 2009). A temperature increase in the source water could hence potentially influence the density and/or quantity of the HSSW produced. Back of the envelope estimates following Nicholls et al. (2009) suggest that a 0.1°C increase would increase the heat flux needed to maintain the HSSW production by 10% – or, for constant heat flux and HSSW production rate, reduce the density of the HSSW by 0.01 kg m$^{-3}$ – but since their estimates are based on relatively cold and highly modified WDW these numbers must be considered an upper limit. The large variability in sea ice formation rates in the region (and hence in HSSW production) has been shown to be largely wind-driven (Hattermann et al., 2021), and the effect of WDW-temperature variability is likely secondary.

## 4.3 Bottom water formation

The outflow of ISW from the FT forms a gravity-driven plume on the continental slope west of the FT sill (Foldvik et al., 2004; Darelius et al., 2009). During its descent towards the deep ocean, the plume water entrains and mixes with the overlying

mWDW, eventually forming Weddell Sea Deep and Bottom Water. As the ratio of ISW to entrained mWDW is on the order of $R_{ISW:mWDW} \simeq 1{:}1.5$ (Foldvik et al., 2004), variability in the properties of the entrained mWDW will translate directly to the properties of the produced bottom water. The seasonality observed in the hydrography of the ambient water at the shelf break is, for example, evident in the plume properties at 1600 m depth (Darelius et al., 2014b). If other factors remain unchanged, we can estimate the imprint of the observed WDW temperature increase on the bottom water temperature following Foldvik et al. (2004): using an outflow temperature of -1.9°C, a mean temperature of the entrained mWDW of 0°C (prior to 2021) and $R_{ISW:mWDW}$ from above suggest that the bottom water produced in 2021 would be about 0.06° warmer than before. At 4000 m depth, the slight increase in temperature would cause the produced bottom water to be (assuming unchanged salinity) 0.01 kg m$^{-3}$ lighter. Due to thermobaricity (the effect of temperature on the compressibility of seawater; cold water is more compressible than warm water) the effect of the temperature increase on density increases with depth. In comparison, the seasonality of the bottom water outflow from the Weddell Sea east of the South Orkney Islands at 4500 m depth has an amplitude of about 0.05°C (Gordon et al., 2010), while the warming trend observed in the bottom waters of the Southern Ocean between the 1990s and 2000s was on the order of 0.03 °C per decade (Purkey and Johnson, 2010) and the range of variability in the mean temperature of the Weddell Sea Deep and Bottom Water at the Greenwich meridian between 1984 and 2008 was on the order of 0.02°C (Fahrbach et al., 2011). Thus, we expect the effect of the local WDW-temperature increase to influence the properties of the AABW produced by the outflow of dense ISW from the FT to be significant, that is, on the same order of magnitude as the seasonality observed downstream (Gordon et al., 2010) and the decadal trends (Purkey and Johnson, 2010; Fahrbach et al., 2011).

## 4.4    What is the origin of the observed local warming?

The high WDW temperatures are anomalous for the upper part of the slope, but water this warm is readily observed both upstream (Fig. 3d) and further offshore (Fig. 3a-c). One can hence easily imagine an "advective" origin of the signal, where the main WDW core is brought closer to the shelf break, or potentially a "mixing" origin, where the mixing and entrainment of colder water into the WDW-core is reduced. The two potential mechanisms will be discussed below.

The continental slope region north of the FT is known to be greatly influenced by continental shelf waves (Jensen et al., 2013; Semper and Darelius, 2017) and local mixing is additionally enhanced by the co-location of near-critical slopes and the critical latitude for the semi-diurnal tide (Fer et al., 2016). Shelf waves can be expected to influence the observed maximum$_{2w}$ temperature at M$_{slope}$ in (at least) two ways; firstly, the waves effectively advect the thermocline up and down the slope (Jensen et al., 2013) and more energetic shelf waves could potentially bring warm water farther up the slope and hence increase the maximum$_{2w}$ temperature. Secondly, however, a larger amplitude in the (largely barotropic, Jensen et al., 2013) velocity signal associated with the waves implies larger bottom shear and more mixing in the bottom boundary layer. This would tend to decrease the maximum$_{2w}$ temperature at M$_{slope}$, since the temperature increases towards the bottom. The time series of M$_{slope}$ temperature and EKE (Fig. 6) suggest that the waves associated with the enhanced peak in EKE during the winter of

2020 could have brought warm water higher up on the slope and caused the observed increase in maximum$_{2w}$ temperature at M$_{slope}$, but it could also be the other way around, i.e. that the EKE is enhanced because the hydrographic conditions are more summer-like and more prone to resonance (Semper and Darelius, 2017). The high maximum$_{2w}$ temperatures during the summers of 2019/20 and 2020/21 could potentially be linked to decreased mixing, as the summer peaks in EKE then are lower than during 2017-2019. We can not rule out that changes in shelf wave activity (through their effect on mixing and/or advection across the slope) are contributing to the changes in observed maximum$_{2w}$ temperature at M$_{slope}$, but it seems unlikely that *all* of the stations from the shallow part of the slope in 2021 should coincide with extreme upslope advection of WDW or that barotropic (depth-independent) waves would contribute to mixing at mid-depth where the temperature maximum is located.

We note two major events that can potentially be linked to the observed temperature increase and the fresh WW layer: the Weddell Sea ice minimum that occurred in 2016 (Turner et al., 2020) and a hydrography transition that occurred in the FT between 2017 and 2018 (Janout et al., 2021).

The sea ice minimum was hypothesized by Ryan et al. (2020) to have caused a longer and warmer than usual inflow of mWDW onto the continental shelf east of the FT in 2017 (observed at ,e.g., at M$_{shelf}$, Fig. 7). The decrease in sea ice concentration also coincides with the onset of a long period with increased inflow of mWDW into the Fimbul Ice Shelf cavity (see Fig. 1 for position) that led to enhanced basal melt rates and stronger cavity circulation (Lauber et al., 2023). The regime shift at Fimbul Ice Shelf is linked to anomalous wind stress forcing, a reduced gradient in Sea Surface Height between the coast and off-shore, and a weaker Antarctic Slope Current (ASC). While barotropic changes in the ASC strength would propagate quickly along the slope (Le Paih et al., 2020; Spence et al., 2017), a freshening signal would need time to develop and to propagate from the source region upstream towards the FT. Graham et al. (2013) estimated (based on results from numerical simulations) that the advective timescale for the fresh anomaly that develops in the eastern Weddell Sea during summer to reach the FT region is on the order of three months. The low sea ice concentration and ice shelf melt along the coast upstream could contribute to the observed fresh WW layer. The results by Bull et al (2021) suggest that there is a potential link between the low salinity of the WW layer and the observed high WDW temperatures. In their numerical simulations, a freshening of the ASC (i.e. a salinity perturbation and positive values of their Slope Current Index, SCI) is associated with increased bottom temperatures above the upper part of the continental slope (their Fig. 8c), consistent with the observations reported on here. Bull et al. (2021) do not, however, further investigate the dynamical link between the ASC freshening and the warming. While we are not able to further quantify the effect, we note that the low salinity of the WW layer increases the stability of the water column. The density difference between the WW and the WDW-core in Fig. 2, for example is 20% larger in 2021 than in the historical mean, and the energy needed to mix the water masses and erode the temperature maximum increases accordingly.

Janout et al. (2021) identified a transition from Ronne-sourced to Berkner-sourced ISW within the FT between 2017 and 2018, as the circulation beneath FRIS intensified due to anomalous wind and sea ice forcing (Hattermann et al., 2021). The system has, however, transitioned between Berkner and Ronne sourced ISW several times within the time span of available observations (Darelius et al., 2014a; Janout et al., 2021), and the change in the density of the FT ISW is relatively small (Janout et al., 2021). While we cannot rule it out, it seems unlikely that the 2017 transition is directly linked to the observed temperature increase on the slope occurring two years later.

## 5  Conclusions

Observations show that the temperature of the WDW core above the upper part of the continental slope in the western Weddell Sea was about 0.1°C warmer in 2021 than previously observed, while the WW-layer above showed a freshening compared to previous observations. Mooring records suggest that a change towards higher temperature maximums occurred late in 2019. While we can not conclude on the origin of the signals, we hypothesize, based on the results by Bull et al. (2021), that the warming and the freshening are linked and that the freshening is associated with the negative sea ice anomaly and increased basal melt rates upstream (Lauber et al., 2023). We show that the observed temperature increase of the WDW can be expected to significantly influence the temperature and density of the AABW formed in the region, potentially affecting the stability of the AABW export from the Weddell Sea (Abrahamsen et al., 2019).

In addition, the results are important to anyone interested in the warm inflow and basal melt rates beneath the Filchner-Ronne Ice Shelf. While the direct effect of the observed temperature increase may be small, these observations redirect the attention to recent modeling work, which highlighted the potential for a regime shift in the area (Hellmer et al., 2012, 2017) and now raises the questions: Are these the first signs of a regime shift in the southern Weddell Sea? How will the situation evolve – will we see rising temperatures on the shelf and increased melt in the years to come? Are the observed changes a local phenomenon, or is there a larger-scale warming/advection of WDW towards shallower isobaths ? How will the newly observed (NASA, 2023) sea ice minimum affect the hydrography at the Weddell Sea continental slope?

Finally, we note that despite its importance, it is becoming increasingly difficult to secure access to the research icebreakers needed to uphold and continue the European efforts to monitor and study the southern Weddell Sea. Long time series and repeat measurements are crucial when addressing climate trends and variability.

*Data availability.*   The CTD data from the COSMUS expedition in 2021 are available in Pangaea: https://doi.org/10.1594/PANGAEA.957614 (Tippenhauer et al., 2023). The historical CTD data analyzed in this study are available for download from pangaea.de, bodc.ac.uk, sea-noe.org, coriolis.eu.org, ewoce.org, ncei.noaa.gov/products/world-ocean-database, and meop.net. CTD data from cruise ES006 (2003) are not searchable but are available from BODC on request (under accession number BAS220012). The doi of individual observational data sets included in the study are listed below.

   Doi of CTD: https://doi.org/10.17882/54012
https://doi.org/10.1594/PANGAEA.61240
https://doi.org/10.1594/PANGAEA.293960
https://doi.org/10.1594/PANGAEA.527233
https://doi.org/10.1594/PANGAEA.527319
https://doi.org/10.1594/PANGAEA.527410
https://doi.org/10.1594/PANGAEA.527497
https://doi.org/10.1594/PANGAEA.527593
https://doi.org/10.1594/PANGAEA.527643

https://doi.org/10.1594/PANGAEA.527812

https://doi.org/10.1594/PANGAEA.733664

https://doi.org/10.1594/PANGAEA.734977

https://doi.org/10.1594/PANGAEA.734988

https://doi.org/10.1594/PANGAEA.735189

https://doi.org/10.1594/PANGAEA.735530

https://doi.org/10.1594/PANGAEA.738489

https://doi.org/10.1594/PANGAEA.742577

https://doi.org/10.1594/PANGAEA.742579

https://doi.org/10.1594/PANGAEA.742581

https://doi.org/10.1594/PANGAEA.756515

https://doi.org/10.1594/PANGAEA.756517

https://doi.org/10.1594/PANGAEA.770000

https://doi.org/10.1594/PANGAEA.772244

https://doi.org/10.1594/PANGAEA.833299

https://doi.org/10.1594/PANGAEA.854148

https://doi.org/10.1594/PANGAEA.859040

https://doi.org/10.1594/PANGAEA.897280

https://doi.org/10.1594/PANGAEA.527646 - 527691

The profiles collected by Argo floats were collected and made freely available by the International Argo Program and the national programs

that contribute to it. (https://argo.ucsd.edu, https://www.ocean-ops.org). The Argo Program is part of the Global Ocean Observing System. The Argo-profiles used in the study were obtained from the World Ocean Database and the Coriolis project (http://www.coriolis.eu.org).

Mooring data are available from www.pangaea.de using the following doi:

https://doi.org/10.1594/PANGAEA.869799

https://doi.org/10.1594/PANGAEA.944430

https://doi.org/10.1594/PANGAEA.875932

https://doi.org/10.1594/PANGAEA.903315

The recent data from $M_{slope}$ are submitted to www.pangaea.de, but until published they can be obtained from the authors on request.

*Author contributions.* ED compiled and analyzed the historical CTD-data, prepared most of the figures, and drafted the ms, VD analyzed the $M_{slope}$ mooring data and made Fig. 6. MJ prepared the temperature records from $M_{shelf}$ and made Fig. 7. ST processed the COSMUS CTD data. All co-authors contributed to the text.

*Competing interests.* The authors have no competing interests.

*Acknowledgements.* This work was supported by the Norwegian Research council through projects 267660 (TOBACO), 231549 (WARM),
and KeyPOCP (328941). This study used samples and data provided by the Alfred Wegener Institute Helmholtz-Center for Polar- and Marine
Research in Bremerhaven (Grant No. AWI-PS124-03). ED is thankful to X. Davis, who kindly provided code and assistance to plot the maps
in Python. We thank K. Heywood and one anonymous reviewer for comments and suggestions that improved an earlier version of the ms.

## Appendix A: Supplementary figures

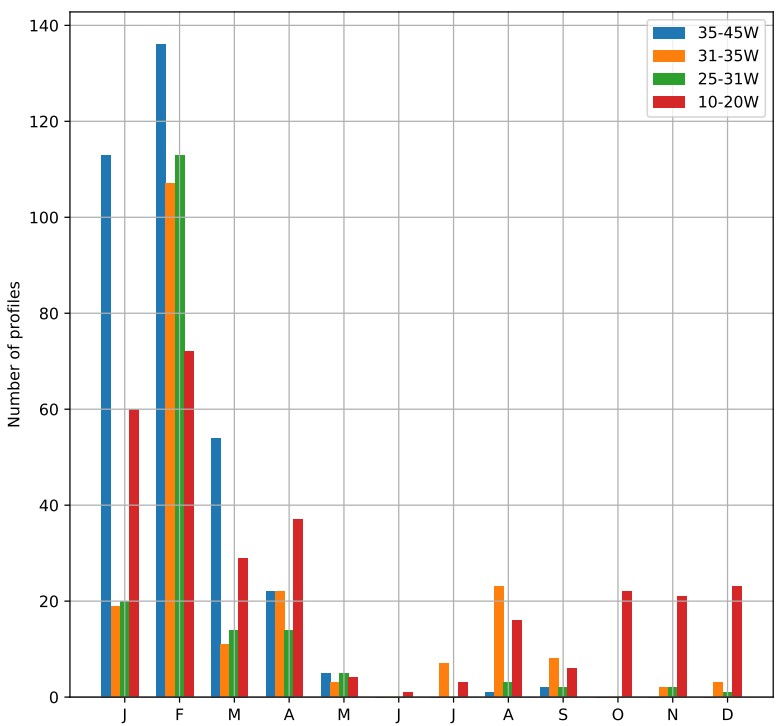

**Figure A1.** Number of profiles per month, per region, included in the study.

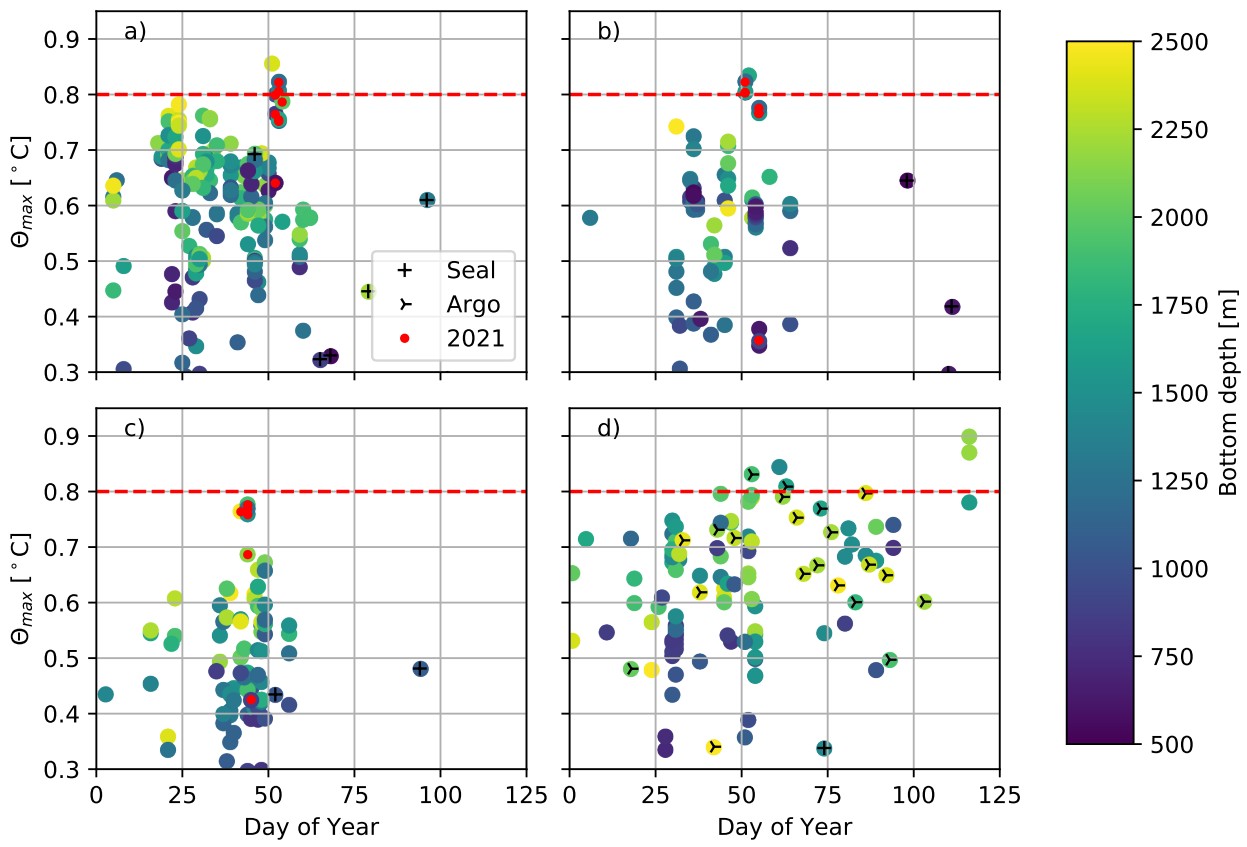

**Figure A2.** Scatter plot of $\Theta_{max}$ versus the day of the year that the profile was obtained, where the color code indicates the bottom depth a) northwest of the FT (35-45W) b) north of the FT (31-35W) c) northeast of FT (25-31W) and d) upstream of the FT region (10-20W). Profiles obtained by instrumented seals, Argo floats, or in 2021 marked according to the legend. The red, dashed line highlights $0.8^{\circ}$C.

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
