# Peer review of "Sudden, local temperature increase above the continental slope in"

_EGUsphere, 2022_

## Referee Comment (RC2)

**Review of "Sudden, local temperature increase above the continental slope in the Southern Weddell Sea, Antarctica" by Elin Darelius, Vår Dundas, Markus Janout and Sandra Tippenhauer**

**Summary comments**

Using a collection of CTDs (historical, COSMUS expedition 2021 and seals) and moorings, this study looks to report on an observed "sudden" increase in the temperature and salinity of Warm Deep Water north of the Filchner Trough in the Weddell Sea starting in 2019. These properties do not appear to be advected downstream towards the ice shelf as downstream observations do not show a signature of the upstream changes. I found this paper interesting and given the recent interest in regime changes of the Filcher-Ronne Ice Shelf, believe that it will also interest the community. In general, it is well written and presented but at times I found the expression and messaging could be tightened up. In particular, a few plot tweaks would make the story much easier to follow. Hence, I recommend acceptance of the paper subject to consideration of some minor, detailed suggestions below.

**Detailed comments**

**Abstract**

Perhaps it's helpful for the reader here to know about the cruise CTDs? (And in general what new observations are presented in this paper.)

**Introduction**

L23. I find 'features' a little awkward

L35. 'equals'? I think you mean is at the same depth?

L47. I think the introduction would have more impact if you close with why we should care about sudden changes in T/S of WDW.

**Data and Methods**

**Conductivity-Temperature-Depth (CTD) profiles**

L55-56. So what is the actual number used? Even just a total in the caption in A1 would be fine.

L64-65. So the water sampling is being used for the OPS? Were any corrections made?

L67-69. I found the words used inconsistent throughout the manuscript. Can you please go through and make all phrasing the same. See also comment below about adding the regions to a Figure please.

**Mooring records**

L73. Add reference to Figure 1?

L73-81. I find it a bit confusing that one moorings is named by its depth and the other by longitude. I guess this is historical (to be consistent with earlier papers)? If not, would it be better to name in terms of their importance to the story? Upstream/downstream etc?

**Results**

L83. 'depth east of the FT'. North-east or even just north? (M31 to me is east)

L95. 'smaller' I would prefer higher or above

L98. 'relatively late' can you be more explicit please. When? Was there anything anomalous, climatically, around this time?

L99. Why is the data here (A2) binned into the regions? And what does the red dots mean?

L103. "The WW is about 0.025 and 0.01 g kg−1 fresher than the mean value and the previously observed minimum, respectively"
I don't understand the 'and' here? Why are there two numbers?
L104. 'Fig4' → 'Fig4a'. Is that unusual? So you are highlighting the absolute number but Figure 4 to my eye, shows ~0.1 variability which is ~10 times more than the amount you're looking to emphasise?

L106. 'Darelius et al, in review' is this allowed in this journal? Perhaps there is a preprint?

L118. I think this is the start of your 'sudden increase', right? So I think it would help the reader if you signpost this paragraph as being important. L124, could be re-phrased and inserted so we come full circle. As I understand it, Figure 2 and 6 are the main results of the paper and could be highlighted as such.

L127. 'average' which is?

L130-138. I don't follow which aspects of the described change are being attributed to changes in EKE? Perhaps some of the text from the discussion should be brought here?

**Discussion**

L139. Perhaps this is a new section (not a sub section)?

L143. Suggest re-phrasing 'maximum_2w' as people just reading this part won't know what that means

L147. This would be easier to follow if each of the three ways were numbered.

L147. Relevant → 'important'?

L151.

> The effects of the observed temperature increase for these **3** processes will be briefly discussed below.

So we have:

1. HSSW

2. FRIS melt
3. Bottom water

Can you please then break up the following text into 3 paragraphs or subsections that address each of these. At the moment I get a little lost, based on the content that is presented, I'm not sure the above is the best way to structure the discussion but if you think that's true, then the earlier material should be made consistent.

L155. It's unusual to me that Figure 7 isn't discussed in the results?

L161. 'Back of the'

L162. Is a 10% change significant? E.g. do we know what the interannual variability of HSSW production is? (I'm hoping for something similar to L180.)

L185. 'Fig' in front of the references

L182. A small re-phrase, drawing together what the reader is supposed to conclude would help I think. Something like: 'Thus, we think that the WDW temperature increase influences bottom water temperature up to…, which is outside the trend of'

L187. So I take it, each of these ideas will now be discussed (following two paragraphs)? Can you signpost it please.

L189-L208. What would be involved in testing how plausible this mechanism is here? It's not my area but perhaps a reduced order model might help corroborate the author's suggestions? Some inspiration could be taken from the mixing estimates given in the Methods in (Meredith et al., 2022) for example.

L209. Here and elsewhere I think it should be: 'Ryan et al (2020)'? See L217 too

L209. For the uninformed reader (this one too, quite a while since I read that paper!).. Are you saying that the time series discussed in Ryan et al. (2020) matches?

**Figures**

*Figure 1.*

Please highlight the four regions of interest on this map (Figure 3-5). Or another 'b' panel if it gets too busy.

Relatedly, much is made of:

> west of the FT (25-31W)
>
> north of the FT (31-35W)
>
> east of FT (35-45W)
>
> upstream of the FT region (10-20W)

Can these longitudes be drawn or highlighted on the map?

Can the mooring and CTD profile be different colors please.

*Figure 2.*

Can the x-axis on a) be a tiny bit wider, would make the dashed line more visible.

Is it worth having a tiny map on this Figure to show where this profile is? You could take the inset off Figure 1 and put it here? Or repeat it without all the green/yellow/black dots.

*Figures 3-5*. Can they please have a discrete colormap.

*Figure 3.* 'Temperature maximum' meaning? Time window? I note this is stated in text for later figures:

> we consider the mean and maximum temperature in two-week-long windows (Fig. 6a-d)

did I miss something?

*Figure 4.* Caption 'west' ← → 'east'

*Figure 5.* There appears to be an artefact on the colormap.

Figure 6. It's a nice Figure but there is a lot going on. I wonder if the black maximum line would be worth showing on Figure 3? It would give the reader a better sense of the findings of the paper, earlier.

The black line on the legend says 'range' but says maximum in the caption. In general, I suggest only explaining things once.

Is it worth showing the raw timeseries? Or is it too noisy without a two week filter? I imagine it's similar to Figure 7. What kind of window filter was used?

References

Meredith, M. P., Inall, M. E., Brearley, J. A., Ehmen, T., Sheen, K., Munday, D., et al. (2022). Internal

tsunamigenesis and ocean mixing driven by glacier calving in Antarctica. *Science Advances*,

*8*(47), eadd0720. https://doi.org/10.1126/sciadv.add0720

---

## Author Comment (AC1)

Dear Prof. Karen Heywood,

Thank you so much for your comments and suggestions – we have updated the manuscript accordingly, and our answers are found below in italic.

Best regards, Elin Darelius & co-authors

Reviewer 1:

I enjoyed reading this paper. It is well written, clearly presented and interesting, with some ingenious analyses. The figures are very good. A particular strength is the analysis of an excellent long-term climatology that has been carefully assembled of ship, seal and float profiles in a region where there have been relatively few studies. It also presents unusually long time series from moorings near the Filchner Ronne Ice Shelf and on the continental shelf and slope. The paper discusses temporal variability, and in particular, possible causes and implications of an occurrence of surprisingly warm WDW in 2021. I found the estimates of the possible impact on AABW very interesting.

I am happy to recommend acceptance of the paper subject to consideration of some minor suggestions that I list below.

The abstract is nice, but it doesn't really do justice to the conclusions you came to about the importance of the observed warming, for example for eventual AABW properties. It would be good to include these implications in the abstract.
*We now mention the potential implications for AABW properties in the abstract.*

L17 I wouldn't say that the Amundsen Sea has a narrow continental shelf – it is hundreds of km from the shelf break to the vulnerable glaciers such as Pine Island and Thwaites.
*No, the shelf is definitely not narrow there. The error is corrected and the text now reads: "where the continental shelf and the ice shelf cavities are flooded by warm Circumpolar Deep Water (CDW)".*

L23 include a reference to moderate melt rates for FRIS?
*We have included a reference to* (Rignot et al., 2013)

L27 typo – you mean tenths not tens of a degree, I think.
*Corrected*

L41 I had to read the bit after the Nicholls reference several times, as it's difficult to make out with the commas and references breaking it up. Try to rephrase to make it easier for readers?
*We have moved the references to the end of the sentence to improve readability.*

L54 I was curious what the source was of the remaining profiles out of the >1000 that are not ship or seals! Later on it's clear that these are profiling floats – I would state that here.
*We now state that the remainder of the floats were collected by profiling Argo-floats.*

Also I recommend giving a reference to MEOP to give due credit to those assembling MEOP data – information about how to cite the data is here https://www.meop.net/database/how-to-cite.html.
*We have included a reference to one of the articles suggested on the meop website.*

Probably there is a similar reference to duly acknowledge the efforts of those assembling the Argo float data set?  Both these data sources should be cited in the text as well as included in the data section at the end.
*We now acknowledge the ARGO-project, and list the data bases from which Argo-data were downloaded.*

L64 what is OTE?  Expand?
*OTE is short for Ocean Test Equipment, the company that produces the bottles, but we decided this information was unnecessary and removed the sentence.*

Caption to Figure 2.  I get that the mentions of 1515 m depth and the 1300-1700m depth range are referring to the sea bed depth, but maybe that needs to be spelt out more clearly?
*We now refer to isobaths rather than depth in the caption.*

L106 and L108 Are these references to the same paper?  In review?  2023?  Reference list says in prep?
*No, these are two different papers. The first one is now published and the reference is updated accordingly. The second one is about to be submitted.*

L116 The in review reference should be updated when available?
*The paper is now published, and the reference is updated*

L118 reference figure 6 here? It takes a while before the reader realises which figure they are meant to be looking at to support this paragraph.  It would be helpful to add references to the relevant figure panel throughout this paragraph to help your readers. E.g. the final sentence, L129,
*We now refer to Fig. 6 in the first sentence of this paragraph and include references to individual panels of Fig. 6 throughout the paragraph.*

Figure 3.  This is a great figure!  Very ingenious. There's a lot of information in these.   I found the order of the 4 panels confusing – would it be easier to follow if the 4 panels progressed W-E or E-W?
*We have re-arranged the order of the panels in Fig 3-5 and corrected the labels (there was apparently some confusion on our side too).*

I wondered whether you had plotted the depth of the Tmax in the same way?  (not necessarily asking for it to be included in the paper, just curious what it might show – it's not quite the same thing as figure 5).

*Here you go – it is basically the same figure, just shifted a bit deeper.*

[Figure]

*Figure 1: Depth of the \Theta-maximum obtained from CTD-profiles a) northwest of FT (35-45W) b) north of the FT (31-35W) c) northeast of the FT (25-31W) as a function of bottom depth. The color indicates the year that the profile was collected.*

L125 I'm being pernickety, but I prefer "decreases" to "drops", here and elsewhere (e.g. l108)
*Corrected*

Caption to Figure 5.  Typo here? The color code is surely the year?
*Corrected*

L136 I think it would be helpful to add some further explanation of why you consider the green line, 24hr, to be tidal EKE.
*This part of the sentence now reads "and ii) the (smaller) wintertime peak in EKE associated with the diurnal tide (B24) is enhanced in 2020 (Fig. 6f, green line) when the mean2w temperature is above the seasonal average"*

L141 I think you should reference the figure showing this result, not the figure showing the locations.
*Corrected*

L141 I would not use the word "now".  We don't know what happened since February 2021, correct?   It might be cold again?  And it's ambiguous when "now" is.
*We agree and we now write that the "temperatures were…"*

Figure 6. What is the red/pink solid line in panels c and d?
*The red line is the seasonal cycle estimated based on the five years of observations. The caption now includes the sentence: "The average maximum temperature (grey dashed line), 0.8◦C (red dashed line), and the mean seasonal cycle based on the five year-long records (red line) are indicated".*

Figure 6. Caption says 12 hours to 2 days but figure legend says 14 hours to 2 days?
*12 hours was a typo. The sentence now reads : ". F) Time series of normalized vertical mid-range EKE (250-500 m depth) at mooring M750 in*
*four frequency bands: 14 hours to 2 days (blue), B12 (orange), B24 (green), and B35 (red)."*

Figure 6. I think you need more explanation in caption, and also in the text, of the band pass filters you chose.  The caption refers to B12, B24 and B35 but these are not defined?
*We have added a sentence describing the filters in section 2.1.1 of the methods: "We estimate the EKE in four frequency bands: around 12h (semi-diurnal tides, B12), around 24h (dirunal tides, B24), around 35h \citep[continental shelf waves, B35,][]{Jensen13}, and 2 days to 14 hours following \cite{Jensen13}. The frequency  bands are marked in Fig. \ref{FIG_M3_mooring_records}e."*
*In the caption of Fig 6 the description of the frequency spectra (panel e) reads: "The shading marks the B35 (red), B24 (green), B12 (orange), and 2 days to 14 hours (blue) frequency bands, color-coded by the legend in panel f)"*

Figure 7.  Is this correct?  These temperatures seem excessively cold, especially for WDW?  They are well below freezing?  Check y axis labelling?
*Welcome to the cold  Weddell Sea, Karen! The y-axis is correct and the water is indeed potentially supercooled. During winter, the continental shelf east of the Filchner Trough is "flooded" by Ice Shelf Water emerging from the Filchner Ronne cavity, see e.g. (Ryan et al., 2017)*

L161 We usually say "back of the envelope".
*Corrected*

L161 suggest (plural estimates)
*Corrected*

L162 Clarify, needed for what?
*We now write "…needed to maintain the HSSW production"*

L168 I think you mean Weddell Sea Deep Water?
*Corrected*

L173 The Foldvik reference needs brackets, also l209 and l217
*Corrected*

L186 the references to the figures need Figure.
*Corrected*

L238 Brackets need moving.
*Corrected*

The paper ends quite abruptly, and I think it would benefit from a bit more discussion of the importance of the results, why they matter (for whom?), and what unanswered questions the work raises?
*We have now extended the conclusion addressing the questions raised by the reviewer.*

The Data availability section doesn't mention float profiles?
*We acknowledge the International Argo Program in the data availability section and state that the Argo-profiles used were obtained from the World Ocean data base and the Coriolis project (without doi).*

References

Hattermann, T., Nicholls, K. W., Hellmer, H. H., Davis, P. E. D., Janout, M. A., Østerhus, S., Schlosser, E., Rohardt, G., & Kanzow, T. (2021). Observed interannual changes beneath Filchner- Ronne Ice Shelf linked to large-scale atmospheric circulation. *Nature Communications*, *2021*, 1–11. https://doi.org/10.1038/s41467-021-23131-x

Purich, A., England, M. H., Cai, W., Sullivan, A., & Durack, P. J. (2018). Impacts of broad-scale surface freshening of the Southern Ocean in a coupled climate model. *Journal of Climate*, *31*(7), 2613–2632. https://doi.org/10.1175/JCLI-D-17-0092.1

Rignot, E., Jacobs, S. S., Mouginot, J., & Scheuchl, B. (2013). Ice-shelf melting around Antarctica. *Science (New York, N.Y.)*, *341*(6143), 266–270. https://doi.org/10.1126/science.1235798

Ryan, S., Hattermann, T., Darelius, E., & Schröder, M. (2017). Seasonal Cycle of Hydrography on the Eastern Shelf of the Filchner Trough, Weddell Sea, Antarctica. *Journal Geophysical Research - Oceans*, *122*. https://doi.org/10.1002/2017JC012916

---

## Author Comment (AC2)

Dear reviewer 2,

Thank you so much for your comments and suggestions – we have updated the manuscript accordingly, and our answers are found below in italic.

Best regards, Elin Darelius & co-authors

Reviewer 2
Review of "Sudden, local temperature increase above the continental slope in the Southern Weddell Sea, Antarctica" by Elin Darelius, Vår Dundas, Markus Janout and Sandra Tippenhauer

Summary comments
Using a collection of CTDs (historical, COSMUS expedition 2021 and seals) and moorings, this study looks to report on an observed "sudden" increase in the temperature and salinity of Warm Deep Water north of the Filchner Trough in the Weddell Sea starting in 2019. These properties do not appear to be advected downstream towards the ice shelf as downstream observations do not show
a signature of the upstream changes. I found this paper interesting and given the recent interest in regime changes of the Filcher-Ronne Ice Shelf, believe that it will also interest the community. In general, it is well written and presented but at times I found the expression and messaging could be tightened up. In particular, a few plot tweaks would make the story much easier to follow. Hence, I recommend acceptance of the paper subject to consideration of some minor, detailed suggestions
below.

Detailed comments
Abstract
Perhaps it's helpful for the reader here to know about the cruise CTDs? (And in general what new observations are presented in this paper.)
*The abstract is updated as suggested.*

Introduction
L23. I find 'features' a little awkward
*We have changed "features" to "experiences"*

L35. 'equals'? I think you mean is at the same depth?
*We have replaced "equals" by "represents"*

L47. I think the introduction would have more impact if you close with why we should care about sudden changes in T/S of WDW.

*We have a included a final statement in the introduction to emphasize why the observed changes matter.*

Data and Methods

Conductivity-Temperature-Depth (CTD) profiles

L55-56. So what is the actual number used? Even just a total in the caption in A1 would be fine.
*We now state the number of profiles actually used (669) in the methods section.*

L64-65. So the water sampling is being used for the OPS? Were any corrections made?
Yes, water samples were used for measurements with the OPS. The difference of salinities measured with the OPS and salinities measured with the CTD sensors are used to determine and correct for a temporal drift of the sensors as well as for a pressure effect, if needed. For this dataset a time dependent correction was applied varying between 0.0028 at the beginning and 0.0063 at the end of the cruise. After the time dependent correction was applied, the pressure dependence was negligible.

L67-69. I found the words used inconsistent throughout the manuscript. Can you please go through and make all phrasing the same. See also comment below about adding the regions to a Figure please.
*We now consistently talk about the area northwest, north and northeast of FT and the area upstream. The longitudes delimiting our regions north of the FT are shown in Fig. 1.*

Mooring records
L73. Add reference to Figure 1?
*We have added a reference to Fig. 1.*

L73-81. I find it a bit confusing that one moorings is named by its depth and the other by longitude. I guess this is historical (to be consistent with earlier papers)? If not, would it be better to name in
terms of their importance to the story? Upstream/downstream etc?
*We have renamed the moorings to $M_{shelf}$ and $M_{slope}$.*
Results
L83. 'depth east of the FT'. North-east or even just north? (M31 to me is east)
*Corrected*

L95. 'smaller' I would prefer higher or above
*We have replaced "smaller" by "shallower"*

L98. 'relatively late' can you be more explicit please. When? Was there anything anomalous, climatically, around this time?
*We now give the dates in the text (8/2 – 15/3), and no there was nothing special, climatically, around this time. The mooring data also indicate that the sudden warming occurred much earlier and persisted.*

L99. Why is the data here (A2) binned into the regions? And what does the red dots mean?
*We chose to display the data in Fig A2 in using the format from Fig. 3-4, and the data are hence binned using the same regions as in these figures. The red dots indicate data from 2021. This is stated in the legend.*

L103. "The WW is about 0.025 and 0.01 g kg−1 fresher than the mean value and the previously observed minimum, respectively" I don't understand the 'and' here? Why are there two numbers?
*The phrase has been reworded.*

L104. 'Fig4' → 'Fig4a'. Is that unusual? So you are highlighting the absolute number but Figure 4 to my eye, shows ~0.1 variability which is ~10 times more than the amount you're looking to emphasise?
*Yes, the variability in the observed WW-salinity is relatively high. This is now mentioned in the ms.*

L106. 'Darelius et al, in review' is this allowed in this journal? Perhaps there is a preprint?
*The paper is now published, and the reference is updated.*

L118. I think this is the start of your 'sudden increase', right? So I think it would help the reader if you signpost this paragraph as being important. L124, could be re-phrased and inserted so we come full circle.
*We have rephrased and introduced the word "sudden" here as suggested.*

As I understand it, Figure 2 and 6 are the main results of the paper and could be highlighted as such.
*We agree that those are the most central figures.*

L127. 'average' which is?
*The "average" is the mean seasonal cycle, i.e. the red line shown in Fig. 6 c-d. This is now stated in the text (and in the red line is explained in the figure caption)*

L130-138. I don't follow which aspects of the described change are being attributed to changes in EKE? Perhaps some of the text from the discussion should be brought here?
*The first sentence in the paragraph now reads: "Energetic shelf waves traveling along the upper part of the slope can potentially affect the temperature observed at Mslope by moving the thermocline up and down and by increasing mixing in the bottom boundary layer."*

Discussion
L139. Perhaps this is a new section (not a sub section)?
*Corrected*

L143. Suggest re-phrasing 'maximum_2w' as people just reading this part won't know what that means

*This and the following sentence now reads: "Mooring records suggest a change to have occurred during 2019, as the maximum temperature in two-week-long windows (maximum2w temperature) observed at the mooring from November 2019 and onward are markedly higher than before (Fig. 6b). During 2020, the seasonal signal in temperature (Semper and Darelius, 2017; Årthun et al., 2012), which is evident in the early part of the record, is not as dominant: the mean temperature during two-week-long windows (mean2w temperature) is markedly higher during this winter compared to previous winters."*

L147. This would be easier to follow if each of the three ways were numbered.
*We have now numbered the three processes.*

L147. Relevant → 'important'?
*We now use "important"*

L151. The effects of the observed temperature increase for these 3 processes will be briefly discussed below.
So we have:
1. HSSW
2. FRIS melt
3. Bottom water

Can you please then break up the following text into 3 paragraphs or subsections that address each of these. At the moment I get a little lost, based on the content that is presented, I'm not sure the above is the best way to structure the discussion but if you think that's true, then the earlier
material should be made consistent.
*We have now introduced subsection to facilitate for the reader.*

L155. It's unusual to me that Figure 7 isn't discussed in the results?
*We now introduce and describe Fig. 7 in the results.*

L161. 'Back of the'
*Corrected*

L162. Is a 10% change significant? E.g. do we know what the interannual variability of HSSW production is? (I'm hoping for something similar to L180.)
*The interannual variability in HSSW production (or at least in sea ice production) is high, and attributed to wind-forcing (Hattermann et al., 2021). This is now stated in the ms, and we expect the effect of WDW-temperature variability to be secondary.*

L185. 'Fig' in front of the references
*Corrected*

L182. A small re-phrase, drawing together what the reader is supposed to conclude would help I think. Something like: 'Thus, we think that the WDW temperature increase influences bottom water temperature up to..., which is outside the trend of'
*We now conclude the subsection as follows: Thus, we expect the effect of the local WDW-temperature increase to influence the properties of the Bottom water produced by the outflow of dense ISW from the FT to be significant, that is, on the same order of magnitude as the seasonality observed downstream \citep{Gordon10} and the decadal trends \citep{Purkey10,Fahrbach11} "*

L187. So I take it, each of these ideas will now be discussed (following two paragraphs)? Can you signpost it please.
*We now state that the two mechanisms will be discussed below.*

L189-L208. What would be involved in testing how plausible this mechanism is here? It's not my area but perhaps a reduced order model might help corroborate the author's suggestions? Some inspiration could be taken from the mixing estimates given in the Methods in (Meredith et al., 2022) for example.

*The density difference between the WW-layer and the core of the WDW in the FT-region is about 20% larger in the profiles from 2021 compared with the historical "mean" profile (Shown in Fig. 2). The amount of energy needed to mix the two water masses (and erode the temperature maximum) has hence increased substantially. This is now stated in the ms . In this case, we see no added value of using the more sophisticated method in the suggested reference (which quantifies the change in stratification/potential energy before/after the ice berg calving)*

*Changes in stratification will, however, impact e.g. the generation and properties of baroclinic instabilities within the ASF/ASC and the strength of tidally generated continental shelf waves (Semper & Darelius, 2017) and hence also the energy that is available for mixing. It is hence beyond the scope of this study to further quantify the effect, but we note that numerical modelling suggests, that at larger scales, a freshening of the upper ocean (from e.g. ice shelf melt) leads to reduced mixing and warming at depth* (Purich et al., 2018).

L209. Here and elsewhere I think it should be: 'Ryan et al (2020)'? See L217 too
*Corrected*

L209. For the uninformed reader (this one too, quite a while since I read that paper!).. Are you saying that the time series discussed in Ryan et al. (2020) matches?
*No, the exceptionally warm and prolonged inflow discussed by Ryan et al occurred in 2017, i.e. prior to the warming we observe on the slope. We now state here that the inflow occurred in 2017, to avoid confusion. The mismatch between the temperature records is discussed in section 4.1.*

Figures

Figure 1.
Please highlight the four regions of interest on this map (Figure 3-5). Or another 'b' panel if it gets too busy.
Relatedly, much is made of:
west of the FT (25-31W)
north of the FT (31-35W)
east of FT (35-45W)
upstream of the FT region (10-20W)
Can these longitudes be drawn or highlighted on the map?
*The longitudes are now shown in Fig 1.*

Can the mooring and CTD profile be different colors please.
*Off course! We now highlight the CTD-profile shown in Fig 2 differently here (white and black).*

Figure 2.
Can the x-axis on a) be a tiny bit wider, would make the dashed line more visible.
*We have increased the limits of (a) slightly.*

Is it worth having a tiny map on this Figure to show where this profile is? You could take the inset off Figure 1 and put it here? Or repeat it without all the green/yellow/black dots.
*We have inserted an inset if Fig. 2.*

Figures 3-5. Can they please have a discrete colormap.
*The colorbar in Fig 3-5 is now discrete.*

Figure 3. 'Temperature maximum' meaning? Time window? I note this is stated in text for later figures: we consider the mean and maximum temperature in two-week-long windows (Fig. 6a-d)
did I miss something?
*The maximum (conservative) temperature plotted in Fig. 3 is the maximum temperature of the WDW-core obtained from CTD-profiles. This is now clearly stated in the caption.*

Figure 4. Caption 'west' ß → 'east'
*Corrected*

Figure 5. There appears to be an artefact on the colormap.
*The strange feature was an empty legend – it has been removed.*

Figure 6. It's a nice Figure but there is a lot going on.
I wonder if the black maximum line would be worth showing on Figure 3?
*The mean two-week temperature maximum is now indicated in Fig 3*

It would give the reader a better sense of the findings of the paper, earlier.

The black line on the legend says 'range' but says maximum in the caption.
*This part of the caption now reads "a-b) is a zoom-in of the highest temperatures at M750 (black)"*

In general, I suggest only explaining things once.
*We have cut some redundancy from the caption of figure 6, which now reads: "a-d) Time series of bottom temperature from Mslope averaged over two-week-long windows, including maximum (black), minimum (grey), mean (blue), the 10 – 90 (filled, grey), and the 25 – 75 (filled, light blue) percentiles. The average maximum temperature (grey dashed line), 0.8◦C (red dashed line), and the seasonal cycle based on the five years (red line) are indicated. a-b) is a zoom-in of the highest temperatures at Mslope. e) Frequency spectra of across-slope velocity (250-500 m depth) at Mslope for winter (dotted line) and summer (black). The shading marks the B35 (red), B24 (green), B12 (orange), and 14 hours to 2 days (blue) frequency bands, color-coded to correspond with panel f). f) Time series of normalized vertical mid-range EKE (250-500 m depth) at mooring Mslope in the same four frequency bands. The colored numbers show the maximum EKE value for each frequency band."*

Is it worth showing the raw timeseries? Or is it too noisy without a two week filter? I imagine it's similar to Figure 7.
*The figure gets quite noisy when the raw data is included, and since we focus on the variability on seasonal and inter-annual timescales, we choose to consider 2-weekly averaged data. The panel below corresponds to Fig. 6c-d), showing the raw data, but does not add information needed for this specific analysis.*

[Figure]

What kind of window filter was used?
*We have used a window length of 2 weeks to estimate the temperature values. The first line of the caption now reads "a-d) Time series of bottom temperature from MSlope averaged over two-week-long windows"*

References
Meredith, M. P., Inall, M. E., Brearley, J. A., Ehmen, T., Sheen, K., Munday, D., et al. (2022). Internal
tsunamigenesis and ocean mixing driven by glacier calving in Antarctica. Science Advances, 8(47), eadd0720. https://doi.org/10.1126/sciadv.add0720

Hattermann, T., Nicholls, K. W., Hellmer, H. H., Davis, P. E. D., Janout, M. A., Østerhus, S., Schlosser, E., Rohardt, G., & Kanzow, T. (2021). Observed interannual changes beneath Filchner- Ronne Ice Shelf linked to large-scale atmospheric circulation. *Nature Communications*, *2021*, 1–11. https://doi.org/10.1038/s41467-021-23131-x

Purich, A., England, M. H., Cai, W., Sullivan, A., & Durack, P. J. (2018). Impacts of broad-scale surface freshening of the Southern Ocean in a coupled climate model. *Journal of Climate*, *31*(7), 2613–2632. https://doi.org/10.1175/JCLI-D-17-0092.1

Rignot, E., Jacobs, S. S., Mouginot, J., & Scheuchl, B. (2013). Ice-shelf melting around Antarctica. *Science (New York, N.Y.)*, *341*(6143), 266–270. https://doi.org/10.1126/science.1235798

Ryan, S., Hattermann, T., Darelius, E., & Schröder, M. (2017). Seasonal Cycle of Hydrography on the Eastern Shelf of the Filchner Trough, Weddell Sea, Antarctica. *Journal Geophysical Research - Oceans*, *122*. https://doi.org/10.1002/2017JC012916

---

## Editor Decision (ED1)

**Minor comments from editor:**

Line 46: you introduced mWDW in Line 45, so also use that to start the following sentence (with MWDW), and remove the 'strongly modified' in Line 47 but rather say by how much, or to what degree that referred to mWDW was cooled.
Line 56: COSMUS expedition (no dash - )
Line 58 …. in the AABW production —> for the AABW production ….
Line 59: I like to ask to change 'dramatically' to something quantifyable e.g. a 10 or 20-fold.
Line 59-60: remove the 2nd 'possibility', e.g. just cut … the possibility … in Line 60.
Line 61: the observed changes in the WDW temperature

Line 65: still add a reference to the ARGO program in the main text body too, eg also include World Ocean Database and the Coriolis project (http://www.coriolis.eu.org) in section 2.1, Line 65

In Figures 3, 4 and 5 add unit degree ° for locations, i.e. °W (in Figure text/legends and captions)

Caption Fig 6: the part: a-b) is a zoom-in of the highest temperatures at Mslope Add to that: a-b) a zoom-in of the temperature range 0.1 to 0.85°C (i.e. the range that is shown there).
End of caption Fig 6: add the unit for EKE (EKE values, which are not normalized) in the caption

Figure 7: Given also the comment of R1, Prof. Dr. Heywood, this mooring temperature record upstream indeed does not show WDW, but mWDW, when -1.6°C<T<0°C (or defined <-0.5°C?)? One can point out in the text when referring to Fig. 7 that one sees intermittently mWDW and ISW there (and not WDW). It will also help the reader to know what the (generally defined) divide is between WDW and mWDW properties in the literature; you could provide that already when introducing the water masses. Also, in subsection 4.1 the title to start with may be mWDW (instead of WDW)? Also, Line 141: is it mWDW or WDW or both? This (if you speak about mWDW or WDW, or both) could be checked and improved generally throughout the ms to clarify (e.g already from the introduction, see also my comment above about Line 46).

**Allowed references:** for final publication, you cannot refer to a paper submitted (or in preparation, or almost submitted). For a final accepted publication in Ocean Science all references need to be published already, accepted for publication, or available as a preprint with a doi. Hence, the reference Darelius et al., 2023b and the sentence where this is referred to needs to be removed or modified.

**Data availability**
Please make sure to make the data from mooring Mslope available when submitting the final ms version and add doi/reference in the Data availability statement.

---

## Author Response (AR2)

Dear editor,

Please find our answers to your comments in *italics* below.

In addition, we identified a sentence where we by mistake had written that the warm inflow contributing to the HSSW production occurs east of FT – this should have been west of the FT (Nicholls et al. 2009) and we have corrected that.

Best regards, Elin Darelius and coauthors

Line 46: you introduced mWDW in Line 45, so also use that to start the following sentence (with MWDW), and remove the 'strongly modi fied' in Line 47 but rather say by how much, or to what degree that referred to mWDW was cooled.
*I start the sentence with "MWDW" as suggested and give the mWDW temperature observed in the Ronne Ice Shelf cavity (-1.4C)*

Line 56: COSMUS expedition (no dash - )
*Corrected*

Line 58 …. in the AABW production —> for the AABW production ….
*Corrected*

Line 59: I like to ask to change 'dramatically' to something quantifyable e.g. a 10 or 20-fold**.**
*I now write that the melt rates may increase by more than an order of magnitude*

Line 59-60: remove the 2nd 'possibility', e.g. just cut … the possibility … in Line 60.
*Corrected*

Line 61: the observed changes in the WDW temperature
*Corrected*

Line 65: still add a reference to the ARGO program in the main text body too, eg also include World Ocean Database and the Coriolis project (http://www.coriolis.eu.org) in section 2.1,
*Corrected*

Line 65 In Figures 3, 4 and 5 add unit degree ° for locations, i.e. °W (in Figure text/legends and captions)
*Corrected*

Caption Fig 6: the part: a-b) is a zoom-in of the highest temperatures at Mslope Add to that: a-b) a zoom-in of the temperature range 0.1 to 0.85°C (i.e. the range that is shown there).
*Corrected*

End of caption Fig 6: add the unit for EKE (EKE values, which are not normalized) in the Caption
*Corrected*

Figure 7: Given also the comment of R1, Prof. Dr. Heywood, this mooring temperature record upstream indeed does not show WDW, but mWDW, when -1.6°C<T<0°C (or defined <-0.5°C?)?
One can point out in the text when referring to Fig. 7 that one sees intermittently mWDW and ISW there (and not WDW). It will also help the reader to know what the (generally defined) divide is between WDW and mWDW properties in the literature; you could provide that already when introducing the water masses.
*The temperature range of WDW (0<T<0.9C,* Gammelsrød et al. 1994) *is now given in the introduction, and we explain when introducing Fig. 7 that the mooring is alternatingly surrounded by mWDW and cold shelf waters / ISW.*

Also, in subsection 4.1 the title to start with may be mWDW (instead of WDW)?
*Corrected*

Also, Line 141: is it
mWDW or WDW or both? This (if you speak about mWDW or WDW, or both) could be checked and improved generally throughout the ms to clarify (e.g already from the introduction, see also my comment above about Line 46).
*Here it would be both mWDW and WDW – we have gone through the ms and corrected our somewhat sloppy use of mWDW/WDW, so that it should now agree with the definition of WDW given in the introduction.*

Allowed references: for final publication, you cannot refer to a paper submitted (or in preparation, or almost submitted). For a final accepted publication in Ocean Science all references need to be published already, accepted for publication, or available as a preprint with a doi. Hence, the reference Darelius et al., 2023b and the sentence where this is referred to needs to be removed or modified.
*The reference to Darelius et al (2023b) has been removed.*

Data availability
Please make sure to make the data from mooring Mslope available when submitting the final ms version and add doi/reference in the Data availability statement.
*The CTD data from 2021 are now available for download from Pangaea* (Tippenhauer et al. 2023) *and the doi is given in the Data availability statement. The mooring data (M_slope) are submitted to Pangaea.de but there is a waiting list (of several months!) and we have not yet obtained a doi. We now write this in the data availability section, and that until published there, the data can be obtained from the authors on request.*

**References**

Gammelsrød, T. et al. 1994. "Distribution of Water Masses on the Continental Shelf in the Southern Weddell Sea." In *The Polar Oceans and Their Role in Shaping the Global Environment, Geophysical Monograph 84*, eds. O.M. Johannessen, R. D. Muench, and J. E. Overland. American Geophysical Union.

Nicholls, K. W. et al. 2009. "Ice-Ocean Processes over the Continental Shelf of the Southern Weddell Sea, Antarctica: A Review." *Reviews of Geophysics* 47: 1–23.

Tippenhauer, S. et al. 2023. "Physical Oceanography Based on Ship CTD during POLARSTERN Cruise PS124." https://doi.org/10.1594/PANGAEA.957614.

---

## Author Response (AR3)

Dear editor,

I have now fixed the missing degree signs in the caption to Fig 2-4, as requested.

Best regards,

Elin Darelius